# 3D Shape Analysis of Powder for Laser Beam Melting by Synchrotron X-ray CT

**Tobias Thiede [1,\*], Tatiana Mishurova [1] , Sergei Evsevleev [1], Itziar Serrano-Munoz [1] , Christian Gollwitzer [1] and Giovanni Bruno [1,2]**

[1]    Bundesanstalt für Materialforschung und -prüfung (BAM), Unter den Eichen 87, 12205 Berlin, Germany;
       tatiana.mishurova@bam.de (T.M.); sergei.evsevleev@bam.de (S.E.); Itziar.Serrano-Munoz@bam.de (I.S.-M.);
       christian.gollwitzer@bam.de (C.G.); giovanni.bruno@bam.de (G.B.)

[2]    Institute of Physics and Astronomy, University of Potsdam, Karl-Liebknecht-Str.24-25,
       14476 Potsdam, Germany

\*    Correspondence: tobias.thiede@bam.de; Tel.: +49-30-8104-4823

**Abstract:** The quality of components made by laser beam melting (LBM) additive manufacturing is naturally influenced by the quality of the powder bed. A packing density <1 and porosity inside the powder particles lead to intrinsic voids in the powder bed. Since the packing density is determined by the particle size and shape distribution, the determination of these properties is of significant interest to assess the printing process. In this work, the size and shape distribution, the amount of the particle's intrinsic porosity, as well as the packing density of micrometric powder used for LBM, have been investigated by means of synchrotron X-ray computed tomography (CT). Two different powder batches were investigated: Ti–6Al–4V produced by plasma atomization and stainless steel 316L produced by gas atomization. Plasma atomization particles were observed to be more spherical in terms of the mean anisotropy compared to particles produced by gas atomization. The two kinds of particles were comparable in size according to the equivalent diameter. The packing density was lower (i.e., the powder bed contained more voids in between particles) for the Ti–6Al–4V particles. The comparison of the tomographic results with laser diffraction, as another particle size measurement technique, proved to be in agreement.

**Keywords:** additive manufacturing; laser beam melting; synchrotron computed tomography; imaging; powder analysis

---

## 1. Introduction

Powder bed additive manufacturing (PB-AM) techniques, such as electron beam melting (EBM) and laser beam melting (LBM), require process control of each manufacturing step [1]. Since the techniques are based on metallic powder, powder characterization is one of the main steps for process optimization. Powder properties, such as heat conduction [2], flowability [3], packing density [4], internal porosity [5], size, and shape [6], may influence the powder bed quality [7].

A variety of powder characteristics (e.g., size distribution, flowability) are certificated by the manufacturer. Usually, the size distribution of powder particles is evaluated by sieving or laser diffraction (LD), which could give a fast and inexpensive overview of the particle size distribution. However, no information about particle shape is available. Different measurement techniques will affect the results of powder characterization [8] and, therefore, hinder their comparison. It has been shown that LD results yield comparable lengths (i.e., maximum diameter) to particle sizes observed by means of X-ray computed tomography (CT) [9]. However, the shape of the particle has an influence on particle size measurement [10], while in cases of LD, particles are assumed to be spherical [11]. The

discussion about particle shape becomes even more critical when recycled powder has to be used during the process, since the mean particle size and shape changes after first use [12–14] and certificates are not reliable any longer. A precise knowledge about the powder is indispensable since the impact on the final part is given for a huge variety of properties [7]. As shown in [15], the usage of realistic powder characteristics during modeling of PB-AM processes is necessary for accurate prediction of porosity and melt pool dimensions.

Since the determination of particle shape is highly complex, it has to be characterized in three dimensions for maximum information [16]. Computed tomography is a common tool for volumetric powder characterization [17]. It allows gaining more statistical information (e.g., number of particles), compared with 2D imaging techniques such as microscopic analysis [14]. Synchrotron X-ray CT (SXCT) is a perfect tool for the characterization of AM powder particles due to its high resolution and fewer image artifacts compared with lab-CT [18]. By applying image processing on CT data [17], different size parameters and shape factors of powder particles can be calculated [19]. One of the critical points for powder characterization is porosity, since it can be transferred into the part and decrease its quality. Chen at el. [20] have shown that porosity in powder particles can depend on the method of powder production as well as on the particle size. Also, the distribution of particles in the powder bed may lead to additional component porosity due to voids between particles. Powders with different size distributions introduce a difference in the powder bed and built part quality [21].

The present work aims to find the correlation of particle size and shape with the packing density of the powder bed (i.e., powder bed quality) by comparing three-dimensional CT measurements with laser diffraction.

## 2. Materials and Methods

### 2.1. Sample Preparation

Two batches of LBM powder particles have been investigated: Ti–6Al–4V produced by plasma atomization (AP&C) and stainless steel 316L produced by gas atomization (SLM Solutions).

For each powder batch, we performed two different sample preparations leading to four samples. First, samples were prepared by filling a glass capillary (internal diameter of 1 mm) with powder to characterize the packing density of each powder batch. In this case, due to the physical contact of powder particles in the glass capillary, the shape and size distribution analysis could not reliably be determined from the CT data. Therefore, each powder was mixed into two-component liquid adhesive (epoxy plus binder) as a second sample preparation, in order to avoid agglomeration of particles for size and shape analysis [9,10]. After solidification of the epoxy matrix, samples with dimensions of 3 mm $\times$ 2 mm $\times$ 10 mm were obtained. The volume fraction of powder in epoxy was around 15% for both samples.

### 2.2. Experiment

The SXCT experiments were carried out at the BAMline at the synchrotron radiation facility BESSY II (Helmholtz Zentrum, Berlin, Germany) [22]. The energy of the monochromatic and parallel beam was varied between 40 keV and 50 keV depending on the material to achieve at least 10% transmission on each sample. An effective pixel size of 0.876 μm was achieved by using a 5× microscope objective (Olympus, Hamburg, Germany) and a CCD-based camera (4000 $\times$ 2760 pixels, PCO, Kelheim, Germany). The distance between the scintillator screen and the object was 10 mm. Three thousand projections over a range of 180° were acquired for each measurement, and each projection had an integration time of 3 s. Ten flat-field images were acquired after every 100 projections, and we corrected the projections with the average of the respective 10 flat-field images. The volume data were reconstructed from the projections by first applying Paganin's phase retrieval algorithm (with $\beta/\delta = 0.027$) [23] and, subsequently, the filtered back-projection algorithm for parallel beam

geometry, using software developed in-house. Based on these volume reconstructions, the packing density, particle size, and particle shape were analyzed.

### 2.3. Packing Density

The packing density was evaluated by using the advanced surface determination module implemented in VG studio MAX 3.2 (Volume Graphics, Heidelberg, Germany). The 3D packing density was defined as the ratio between the number of material voxels and the total number of voxels within the volume. It was calculated for variously sized and positioned regions of interest (ROI) within the sample, in order to assess the statistical uncertainty, see Figure 1.

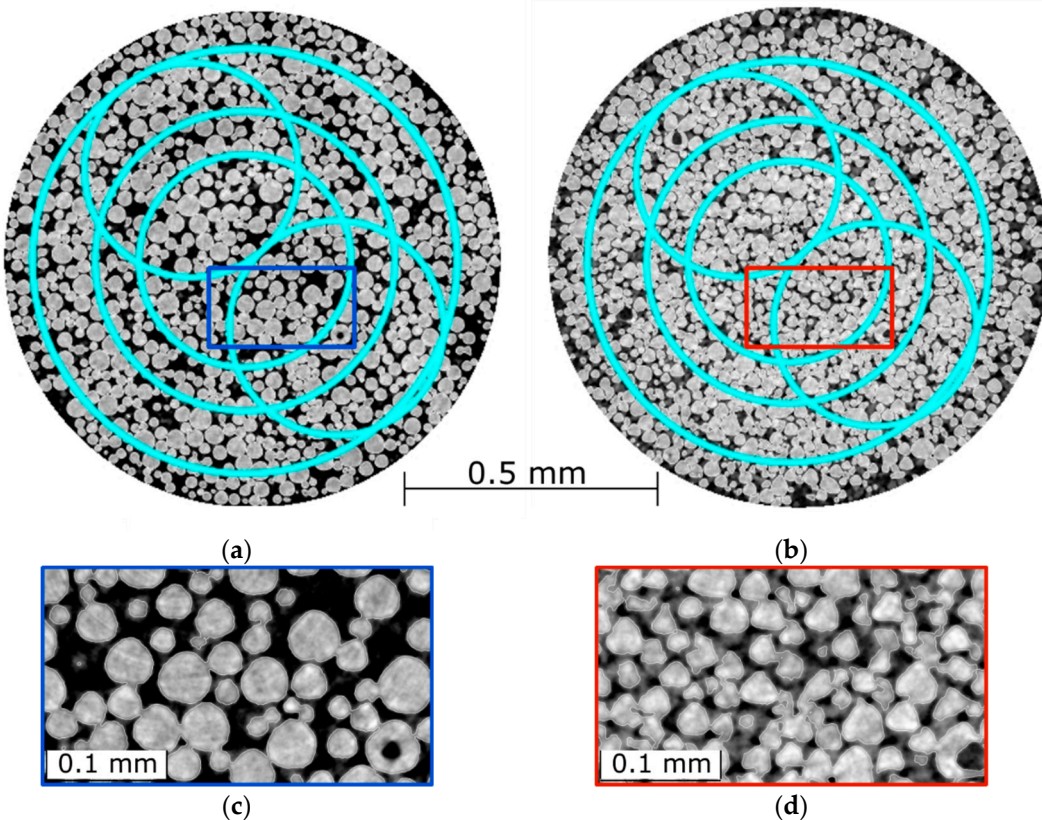

**Figure 1.** (**a**) Ti–6Al–4V and (**b**) steel 316L laser beam melting (LBM) powder in glass capillary. The analyzed regions of interest (ROI) (turquoise, VG Studio MAX 3.2) are also shown. Enlarged view for Ti–6Al–4V (blue box) (**c**) and for steel 316L (red box) (**d**).

### 2.4. Particle Segmentation and Shape Analysis

The shape analysis required segmented data with all particles being separated from each other. The particle segmentation workflow was performed by using ImageJ [24]. Prior to segmentation, noise reduction was required, which was done by application of a bilateral filter [25]. The bilateral filter was defined as a weighted average of pixels. It took into account not only the spatial distance of pixels but also the variation in their grey values (i.e., similarity in the range). Therefore, the bilateral filter allowed for the suppression of noise while preserving the edges of particles. The following filter parameters provided the optimal combination of denoising and edge preservation: a spatial distance kernel of 7 pixels and a grey value range kernel of 50. Figure 2b shows a slice of the filtered reconstruction of the CT data of the steel 316L powder epoxy sample.

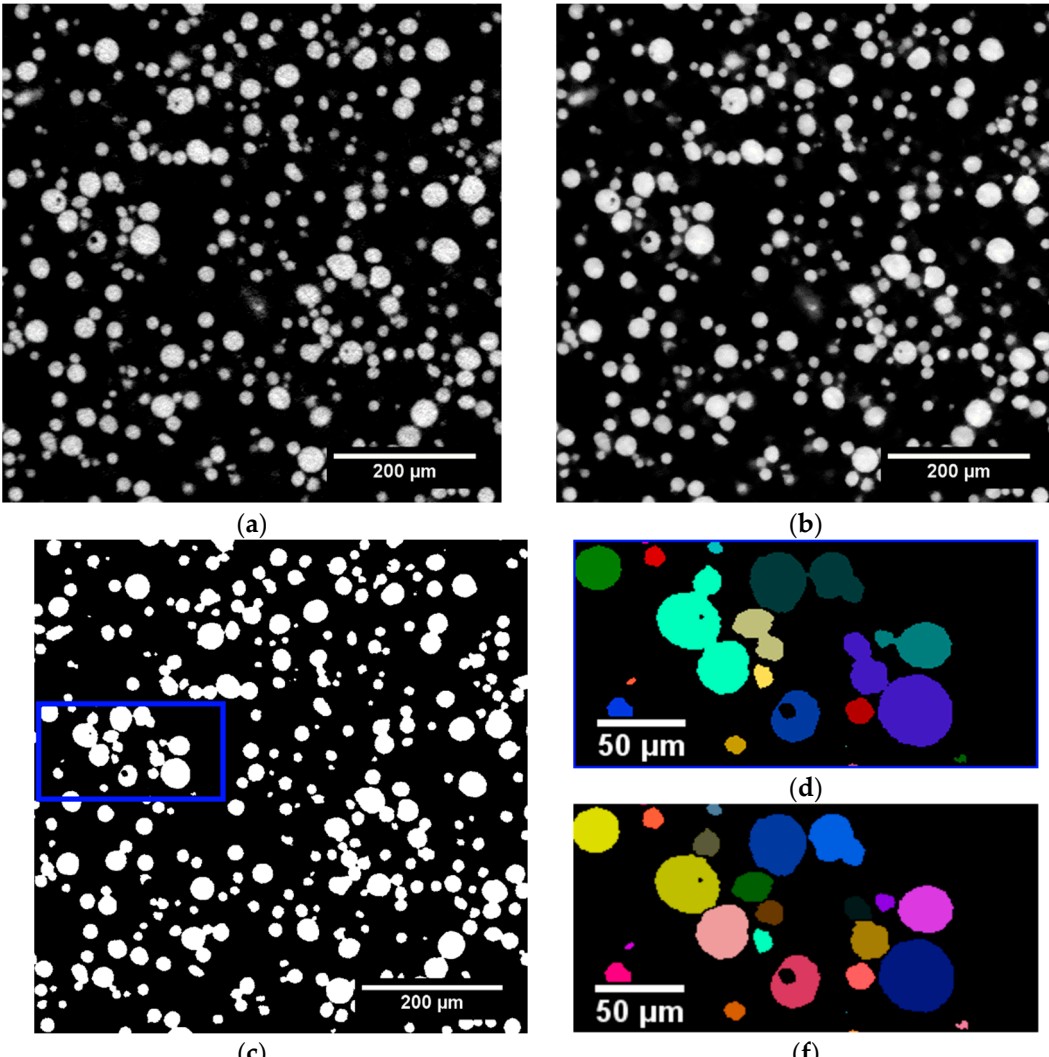

**Figure 2.** Illustration of particle segmentation workflow for steel 316L powder in epoxy: (**a**) Computed tomography (CT) slice of reconstructed data; (**b**) CT slice of bilateral filtered data; (**c**) CT slice of binarized data; (**d**) the enlarged view of the marked area in (**c**) (particles are labeled for better visualization); (**e**) the result of watershed segmentation.

The second step of the segmentation workflow was the determination of a 3D particle mask and the separation of particles from the surrounding background. The automatic thresholding implemented in ImageJ was employed. The final threshold value was defined by the iterative procedure based on the ISO data algorithm, which had first been introduced in [26]. Figure 2c shows the resulting binary image after global thresholding. It can be seen, that even in the samples with powder diluted in the epoxy matrix, some particles remained attached together, see Figure 2b,c.

A widely used approach for particle recognition and fragmentation was the watershed segmentation algorithm. An advanced watershed approach for 3D particle recognition (implemented plugin in ImageJ) that tolerated particle concavities was applied [27], see Figure 2d. The idea, which differs from the conventional watershed, was an additional controlling parameter k for limitation of the fragmentation $0 < k < 1$, where $k = 1$ corresponded to conventional watershed fragmentation and $k = 0$ would lead to no fragmentation [27]. This helps to avoid the over-partitioning, which occasionally takes place in conventional watershed algorithm, for which any concavity of the particle surface leads to separation. For the analyzed datasets, the best segmentation result with the minimum amount of improper fragmentations was achieved with k = 0.7.

A dataset with a size of 1.752 mm × 1.752 mm × 0.876 mm (2000 × 2000 × 1000 voxels) was analyzed.

After particle segmentation, the particle size and shape were analyzed by VG Studio Max 3.2 and Amira ZIB Edition 2017.47. Various features such as sphericity, anisotropy, and principal geometrical components were determined.

A principal component analysis (PCA) [28] was conducted for each segmented particle. The PCA provided the eigenvalues of the covariance matrix $\lambda_1$, $\lambda_2$, and $\lambda_3$ for each particle. The eigenvalues $\lambda_1$, $\lambda_2$, and $\lambda_3$ correlated with the diameters along the direction of the three eigenvectors of the covariance matrix. The eigenvectors of the covariance matrix physically corresponded to the axes of inertia of the particles. The correlation between eigenvalue and diameter along each eigenvector was found by analysis of known simulated ellipsoids, see Equation (1). We simulated variously sized voxel-based ellipsoids, each in a binarized volume of 256 × 256 × 256 voxels, and estimated the proportionality factor between eigenvalues and size along the respective direction to be a factor of 0.225.

$$\text{length} = \frac{\sqrt{\lambda_1}}{0.225}; \ \text{width} = \frac{\sqrt{\lambda_2}}{0.225}; \ \text{height} = \frac{\sqrt{\lambda_3}}{0.225}. \tag{1}$$

Since these three respectively orthogonal diameters can be understood as the bounding box around the particle, see Figure 3 [9,10], we named them:

1.　length = largest diameter (corresponds to $\lambda_1$).
2.　width = medium diameter (corresponds to $\lambda_2$).
3.　height = smallest diameter (corresponds to $\lambda_3$).

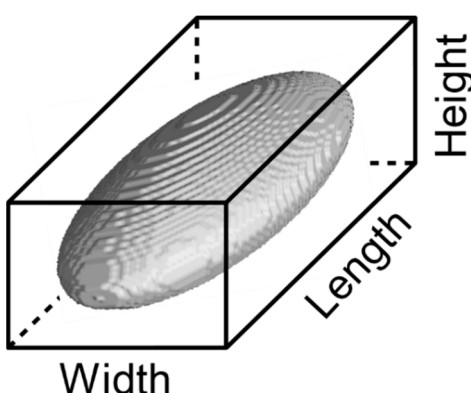

**Figure 3.** Sketched dimensions for a simulated ellipsoid.

The sphericity S was described by the ratio between the surface area of a sphere with the same volume as the particle, and the surface area of the particle itself. The method was only meaningful, if the surface of the particle was not a voxel-based (described as 'discrete' in the software Amira (2019.03, ZIB, Berlin, Germany) surface but a smoothed (described as 'continuous' in the software Amira) surface.

$$S = \frac{\sqrt[3]{36\pi \cdot Volume^2_{particle}}}{Surface_{particle}}; \ S_{sphere} = 1. \tag{2}$$

The anisotropy, *A*, was calculated according to the following equation:

$$A = 1 - \frac{\lambda_3}{\lambda_1}; \ A_{sphere} = 0, \tag{3}$$

where $\lambda_1$ and $\lambda_3$ are the largest and smallest eigenvalue of the PCA, respectively.

## 3. Results and Discussion

### 3.1. Particle Size

The size of a particle can be described by different variables. The volume was the most intuitive one. The equivalent diameter (i.e., the diameter of a sphere with the same volume as the particle) was often used to give an impression about the dimension of the particle. The distribution of the equivalent diameter of both powder batches is presented in Figure 4. Additionally, we compared the results of our powder particle size analysis with the powder certificate presented by the manufacturer, see Table 1. $D_{10}$, $D_{50}$, and $D_{90}$ represented the value of the cumulative frequency at 0.1, 0.5, and 0.9, respectively. The mean and the standard deviation of the log–normal function fitted against the size distribution yielded complementary information (the position and the spread of the distribution) to $D_{10}$, $D_{50}$, and $D_{90}$. The mean size $\mu$ was 2 µm bigger than $D_{50}$ for both powder batches. The fact that the difference $D_{90}$–$D_{50}$ was always bigger than $D_{50}$–$D_{10}$ supported the asymmetry of the distributions, which was found by fit of the log–normal distribution.

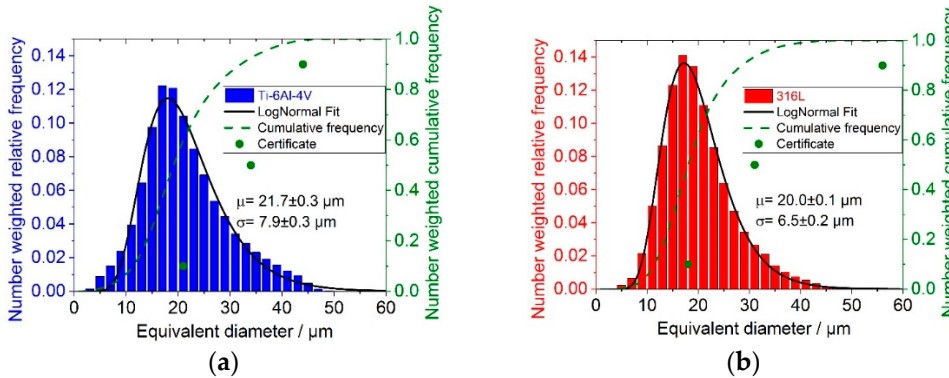

**Figure 4.** Particle size distribution and cumulative frequency (dashed line) of (**a**) Ti–6Al–4V powder batch and (**b**) 316L steel powder batch. Both distributions are fitted with a log–normal function (black line). The certificated value for $D_{10}$, $D_{50}$, and $D_{90}$ are presented as green dots.

**Table 1.** Comparison of volume-weighted powder particle size distributions obtained by laser diffraction (LD) and by CT.

| Material | Measurement | $D_{10}$/µm | $D_{50}$/µm | $D_{90}$/µm |
|---|---|---|---|---|
| | Certificate | 18 | 31 | 56 |
| | Length | 18.9 | 29.8 | 45.3 |
| 316L | Equivalent diameter | 17.2 | 26.5 | 39.4 |
| | Width | 17.1 | 26.4 | 39.3 |
| | Height | 15.5 | 24.2 | 36.2 |
| | Certificate | 21 | 34 | 44 |
| | Length | 19.6 | 31.2 | 43.6 |
| Ti–6Al–4V | Equivalent diameter | 18.6 | 30.0 | 42.0 |
| | Width | 18.6 | 30.0 | 41.9 |
| | Height | 17.8 | 29.0 | 40.7 |

An additional method for size analysis was the PCA of grey value distribution of each particle [28]. The resulting sizes according to Equation (1) are summarized in Table 1. The equivalent diameter and the PCA diameter were complementary, since the equivalent diameter was always smaller than length, but larger than height. Both the equivalent diameter and the PCA results showed that the 316L steel powder batch had smaller particles and a narrower size distribution compared to Ti–6Al–4V.

The values given for $D_{10}$, $D_{50}$, *and* $D_{90}$ in the powder certificate (determined by LD) could not be directly compared to the CT results. Without data treatment, the particle size distribution based on CT results was a number-weighted size distribution, since the particles were labeled inherently during the segmentation process (Figure 4), whereas LD provided volume-weighted size distributions according to ASTM B 822 [29] and ISO 13320 [30]. If this was not taken into account it would have led to large discrepancies. Therefore, we also evaluated the volume, instead of number of particles, from the CT measurements. The cumulative frequency of the volume-weighted size distribution for the length of the particles is shown in Figure 5 for both powder batches.

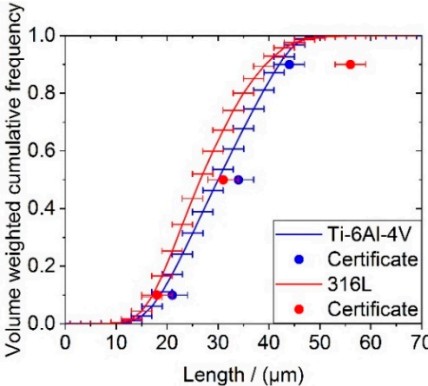

**Figure 5.** Volume weighted cumulative frequency for Ti–6Al–4V (blue) and 316L (**red**) based on CT analysis in comparison with the certificated values for $D_{10}$, $D_{50}$, *and* $D_{90}$ (**filled circles**).

The certificated values $D_{10}$, $D_{50}$, *and* $D_{90}$ agreed better for Ti–6Al–4V compared to steel 316L. This might be due to the higher anisotropy of Ti–6Al–4V particles compared to steel 316L. For the LD signal evaluation, all particles are assumed to be perfect spheres. The uncertainty contribution to the LD-measured diameter caused by this assumption was estimated using a simulation. The scattering from an ensemble of ellipsoidal particles with a log–normal size distribution according to the CT result and an aspect ratio of 0.7 (the maximum aspect ratio observed in the CT data) was simulated using the small-angle scattering form factor [31]. A model of log–normal distributed spheres was least squares-adjusted to the resulting scattering curve, to simulate the data evaluation of a commercial LD device using Fraunhofer diffraction [32]. The deviation of the mean diameter from the nominal diameter amounts to 2–3 µm for a particle ensemble with $A < 0.7$. An additional uncertainty contribution explaining the difference between CT and LD is the surface determination of CT grey-value data. The error can be estimated to be one voxel (0.876 µm), which leads to an error of 2 µm for the diameter. Except $D_{90}$ for steel 316L, all values lie within the error. An artifact of the applied particle segmentation was the segmentation of truly sintered particles. However, this oversegmentation affects only a small portion of particles as it can be observed by means of optical microscopy of LBM powder [7,33,34]. The amount of false particle segmentation should be vanishing within the statistics of the analyzed number of particles (70,000 for steel 316L and 50,000 for Ti–6Al–4V).

### 3.2. Particle Shape

The particle shape was analyzed by two different shape parameters: sphericity and anisotropy, see Figure 6. The anisotropy showed a larger difference between 316L and Ti–6Al–4V, while the sphericity was less different (Figure 6).

For a perfectly convex particle (i.e., an ellipsoid) close to a sphere, the sphericity was only slightly influenced by the aspect ratio. The length of three principal axes was statistically distributed between 15 and 45 µm (see Table 1). According to Equation (2), in such conditions the sphericity changes from 1 to 0.92 for a perfect ellipsoid. However, we observed sphericity values lower than 0.92. This provided information about the concavity of particles (induced, e.g., by open porosity). In our case, the powder

particles showed high sphericity and, hence, a high degree of convexity, which justifies the description of the particles as ellipsoids (i.e., the PCA).

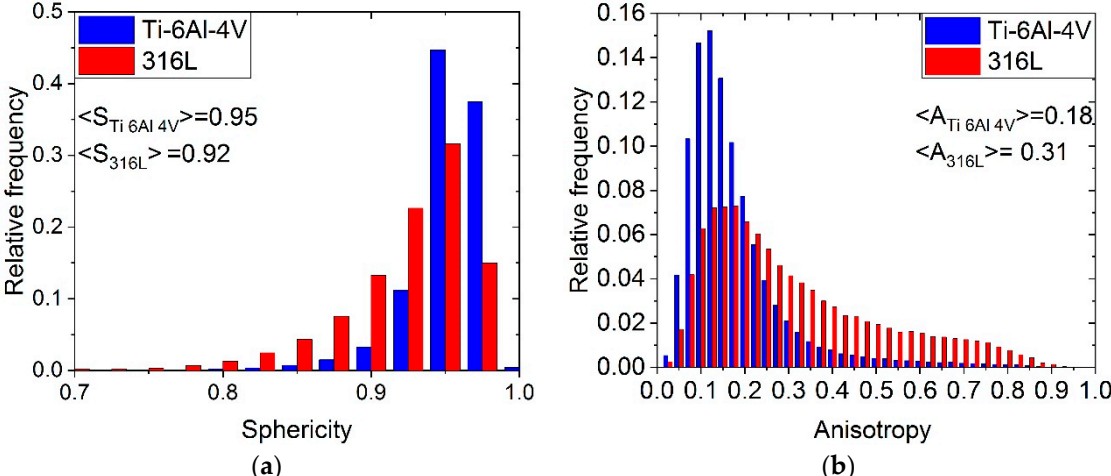

**Figure 6.** (**a**) The sphericity and (**b**) the anisotropy distribution of Ti–6Al–4V powder particles (**blue**) and 316L steel powder particles (**red**). The arithmetic means are presented as an inset.

The anisotropy distribution of 316L (Figure 6b), which is based on PCA, showed a shoulder for high anisotropy ($A \approx 0.8$), since the anisotropy was based on the particle aspect ratio. Compared to the sphericity, the anisotropy showed a larger difference between the two powders. This allowed for the conclusion that the two different production techniques (plasma and gas atomization) led to comparable convexity, but gas atomization yielded a lower aspect ratio in this case.

However, the two methods to describe the particle shape led to the same qualitative result: the Ti–6Al–4V powder produced by plasma atomization was more spherical than the 316L powder produced by a gas atomization process.

### 3.3. Packing Density

The packing density (PD) was assessed for the powder filled into glass capillaries. The evaluation of the five ROI described in the Methods section (see Figure 1) led to the following two packing densities (PDs):

1.　$PD_{Ti-6Al-4V} = 0.561 \pm 0.003$.
2.　$PD_{316L} = 0.576 \pm 0.004$.

The error represented the statistics within the glass capillary. The slight difference between the two powders was statistically representable as it was well above the error bar. The influence of the container size on the packing density of the particles was described in [35,36]. The publications have been written for mono- and bi-dispersed powders, respectively. Since our powder was neither mono- nor bi-disperse, we took the mean size (20 μm, Figure 4) to verify if we are, on average, within the right container size (1 mm). The ratio (1000 μm/20 μm) corresponds to the plateau of maximum theoretical packing density, where the sample size was statistically representable [35]. Hettiarachchi et al. confirmed that the container wall effect can be neglected when the ratio of particle size and container size is less than 0.1 [36]. A container of 1 mm would allow a maximum particle diameter of 100 μm. All of our particles were well smaller than 100 μm. Therefore, the chosen sample geometry can be regarded as a representative volume.

The powder particle porosity (i.e., the closed voids within a particle) did not influence the packing density analysis, since the particle porosity was the same (0.03%) for both powder batches and was negligibly small.

Although the packing density was nearly the same for both powder batches, the voids between the powder particles were different. Figure 6a indicates a shift of interparticle distance of Ti–6Al–4V towards larger voids. The interparticle distance (see Figure 7c) was calculated by means of a 3D distance map on a binarized inverted 3D volume (see Figure 7b). The more round-shaped Ti–6Al–4V particles induced less, but bigger, voids between the particles. The influence of this different void distribution on the powder bed quality will need further investigation.

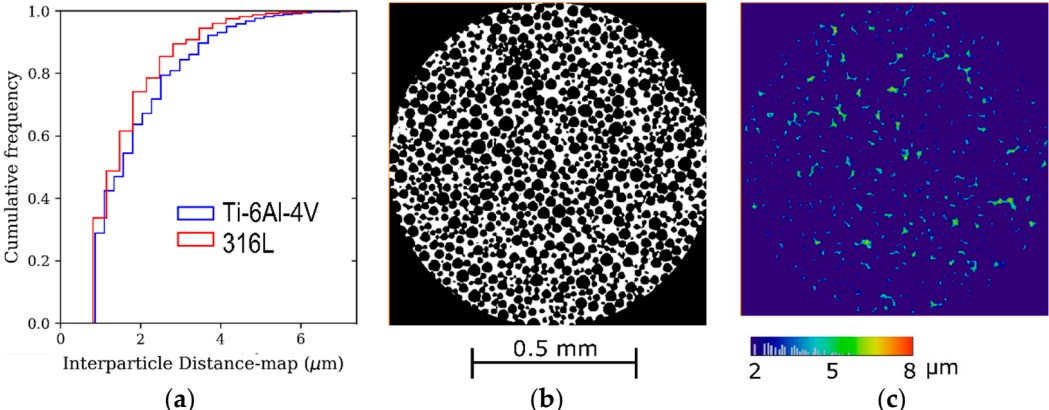

**Figure 7.** (**a**) Cumulative frequency of interparticle distance evaluated by 3D distance map for powder in capillary; (**b**) 2D slice of binarized and inversed volume; (**c**) the 3D distance transformation of (**b**).

The goal of this work was to correlate the packing density with the size and shape of the powder particles, in order to understand the powder bed quality. As shown in [37], a higher packing density correlated with a higher density of produced parts. Therefore, the packing density of the powder bed was an important process parameter for LBM process. The maximum packing density for ordered monodisperse spheres is known to be 0.74. For unordered monodisperse spheres, a packing density of 0.635 has been reported [38]. In this work, polydisperse powder distributions were discussed. Pednekar et al. presented statistically equivalent bidisperse distributions for log–normal distributions [39]. We estimated the statistically equivalent bidisperse distribution for our polydisperse distributions. This transformation led to representative radii ratios of $D_S/D_L$ for 316L and $D_S/D_L = 0.51$ for Ti–6Al–4V, with $D_S$ and $D_L$ being the diameter of smaller and the larger particles, respectively. They correlate with a slightly higher packing density for 316L steel. In general, this shows that both radii ratios are not small enough to allow an increase of packing density compared to the nominal value of monodisperse powder (0.635). These observations qualitatively match our observed packing densities of 0.56 and 0.58.

Steel 316L was produced by gas atomization and Ti–6Al–4V by plasma atomization, which led to a different shape in terms of anisotropy of the powder particles (Figure 6b). However, the PD of steel 316L and Ti–6Al–4V showed only a small difference. Therefore, the different shape of the powder particles (Figure 6) does not have a strong influence on the packing density and, presumably, on the powder bed quality.

## 4. Conclusions

We presented a robust workflow for 3D particle size and shape analysis by means of synchrotron CT with sufficient particle statistics.

We showed that the dimensions (i.e., length, width, height) of the powder particles, which represent a bounding box around each particle, are in good agreement with the equivalent diameter representing the volume only.

A detailed knowledge regarding the particle size distribution could be used for optimization of the layer thickness during LBM process, as has been recommended in [40]: the process layer thickness

should not exceed the $D_{90}$ value. Also, it has been shown by simulations that smaller particles may compensate for defects in the powder bed [41,42]. This means that experimentally gained information about powder size and shape distribution can be additionally used as an input for powder bed simulation. The packing density and the interparticle distance influence not only the intrinsic voids but also physical properties, such as the thermal conductivity of the powder bed. According to these results, we will use the length, width, and height for size modeling and the anisotropy for simple shape modeling.

**Author Contributions:** Conceptualization, T.T., T.M., and C.G.; Formal analysis, T.T., T.M., S.E., I.S.-M., and C.G.; investigation, T.T., T.M., S.E., I.S.-M., and C.G.; Resources, T.T. and T.M.; Software, S.E., I.S.-M., and C.G.; Supervision, T.T.; Visualization, T.T., S.E., and I.S.-M.; Writing—original draft preparation, T.T., T.M., and S.E.; Writing—review and editing, T.T., T.M., S.E., I.S.-M., C.G., and G.B.

**Funding:** This research received no external funding.

**Acknowledgments:** We acknowledge Gunther Mohr (BAM) and Katia Artzt and Jan Haubrich (DLR Institute of Materials Research) for donations of powder materials.

**Conflicts of Interest:** The authors declare no conflict of interest.

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
