# Peer review of "3D Shape Analysis of Powder for Laser Beam Melting by Synchrotron X-ray CT"

_qubs, doi:10.3390/qubs3010003_

Round 1
Reviewer 1 Report
A good paper on an interesting subject. There are a few places where the discussion is not clear enough, which should be addressed in order to make the paper more readable and useful. Minor revisions are needed. My detailed comments follow.
Lines 49-50 Not describing particle shape as a scalar really has nothing to do with 2D vs 3D. The shape of a random particle cannot be completely described by any finite set of numbers. One chooses a finite set of numbers that can adequately approximate the shape. Of course, this must be done in 3D to have any hope of being accurate, unless of course the shapes are quite simple, like spheres. There are exact relations between 2D projections and 3D reality for ellipsoids (see Vickers, Powder Technology).
Lines 72-75 What was the approximate volume fraction for the powders in epoxy? This technique was first used in refs. 8 and 9.
Figure 2d: Breaking apart the green multi-particle with the watershed algorithm may lead to inaccuracies, since this could really and probably is a true multi-particle. Later on the authors state that this could happen but would not make a difference, since the number of these particles was small. The authors present no evidence for this claim. I think there could be a lot of such multi-particles, with a significant volume fraction.
Equations 1 and 2: I would prefer standard mathematical notation for equations.
Line 136 What do the lambdas mean, physically, for a given particle? Are they the axes of some kind of equivalent ellipsoid? If so, how is the equivalency made – volume, surface area, etc.?
Lines 155-169 The definition of the length, width, and height is not clear at all. Where does the factor 0.225 come from? What is the meaning of eq. 3? Refs. 8 and 9 contain a definition of length, width, and thickness – are these the same as in this paper? Table 1 gives only average values – how about distributions? In general, the definition of length, width, and height needs to be greatly clarified, and their relation to the PCA parameters need to be better described.
Line 171 I don’t see why the CT results are “naturally” number-based – data can be presented in any form one desires.
Figure 4: I assume this data is the CT data – it should be so labelled.
Lines 185-195 I like the ellipsoid simulation. An older and similar reference that should be considered is:
H. Mühlenweg, E.D. Hirleman, Laser diffraction spectroscopy: influence of particle shape and a shape adaptation technique, Particle & Particle, Systems Characterization 15 (4) (1998) 163–169.
Lines 197-198 Evidence for this statement?
Line 203 “Sphericity is a measure of convexity…” I don’t think that this is a true statement at all. The authors need to better justify this statement, if it is true, which I doubt. “For a perfectly convex particle (i.e., an ellipsoid) – this is a sentence fragment and I am not sure what it means.
Line 204 “The sphericity is only slightly influenced by the aspect ratio…” Again, I don’t think that this is a true statement. For ellipsoids that are close to spheres, this could be the case, but for very prolate or very oblate ellipsoids, this will not be true. It is easy to check – just find a good approximate formula for the surface area of an ellipsoid and use it to evaluate eq. 1.
Line 210 What does “compactness” mean in this context?
Lines 220-221 Knowing the uncertainty in segmentation, and how sensitive the volume fraction of packed powder is on this segmentation, I find it hard to believe, without further justification, that the uncertainties for the PD are so small.
Lines 222-228 First, there has been an enormous amount of work on particle packing since 1961, the date of Ref. 30. Second, the powder in this paper is nowhere near being monodisperse, so some justification is needed to be able to use the results of Ref. 30.
Lines 229-231 Is this value of porosity, 0.03 %, averaged over all particles or over particles that have porosity? There can be a big difference between these two ways of averaging. There can be relatively few particles that have pores, but if there are a lot of pores per particle, there can be a significant amount of defects introduced into the built part.
Lines 245-250 An 0.74 packing fraction is for an ordered packing of monosize spheres. How did the authors calculate the value of 0.41 for their powder, since both powders are nowhere near being bi-disperse. They must have made an approximation, which the reader should be able to see and therefore evaluate. I note that Torquato’s (Random Composites) book has a lot of information in it about packing, and there has been a lot of work since the publication of that book.
Lines 252-255 But I thought the authors previously stated that the differences in PD for the two powders was significant, now they say the difference is negligible. Real but negligible? I would say the error bars on the PDs should be larger, so that that no difference could be determined in the PD.
Line 269 I again note that length, width, and height have not been clearly defined in this paper.
Author Response
Response letter to the reviews of
3D Shape Analysis of Powder for Laser Beam Melting by Synchrotron X-ray CT
By Tobias Thiede, Tatiana Mishurova, Sergei Evsevleev, Itziar Serrano Munoz, Christian Gollwitzer and Giovanni Bruno
Dear Editor,
We thank you and the reviewers for the thorough and constructive review of our manuscript. We appreciate the time they have spent to suggest us corrections and items of discussion. Indeed, we carefully considered all comments, and changed the manuscript accordingly.
In the following we comment in detail to all points raised.
We trust the manuscript has greatly improved and will be of great interest to the readers of QuBS.
On the behalf of all other authors, I am sending our best regards
Tobias Thiede

Reviewer 2 Report
The manuscript reflected significant effort by the authors. However, significant revision is recommended before publication could be considered. The following issues should be addressed:
(1) The introduction section should be revised to show the significance of the work, and the conclusion section should be revised to demonstrate the connection between powder size distribution and 3D-print quality. Otherwise, it appears that the work is just running through a X-ray CT and image analysis routine for a particular batch of powder from a particular supplier.
(2) Packing density from X-ray CT imaging should be compared with the actual value estimated by the weight and volume of the powder and epoxy used for sample preparation, or other measurements.
(3) More details about the sample should be reported. Such as the amount of the powder and epoxy used, mixing method etc. The sample should be a 3D shape. However, it is reported to be a 2D rectangle in line 75 on page 2.
(4) In line 79 on page 2, what is the "electron density" of each sample? How are they obtained? What is the quantitative relation between this "electron density" and the X-ray beam energy?
(5) The authors may want to pay more attention to the X-ray imaging process. Such as, flat-field and dark-field images are often acquired to compensate background artefacts. It sounds that the X-ray image detector is a bit inefficient that 3seconds of exposure time is required. Sample to detector distance is not reported.
(6) Parameters for the phase-retrieval algorithm is missing. Ring artefact is not handled although it is obvious on the images.
(7) What is the relation between the VG studio "advanced surface determination" and the ImageJ segregation? How consistent are the two different approaches?
(8) SEM/EDX images of powder cross sections are recommended. The particles show obvious density variations.
(9) It is known that the CT reconstructed particle density is affected by the point-spread-function (POI) effect. Using the image analysis approach as reported in the paper, small particles appear larger than the actual size while the density appear lower than the actual linear absorption coefficient. Very small particles would not be detected. An effect mass concept, that is the volume times density, might be a more accurate representation. The authors may like to do a keyword search on "data-constrained modelling" and pay attention to partial volume effect.
(10) All equations need to be re-written using common mathematical convention, and all symbols should be described. The authors are encouraged to ask someone to go through the equations who has an equivalence of undergraduate mathematical training. In additional to the equations and as another example, lamda_{1,2,3} is different from {lambda_1, lambda_2, lambda_3}.
Author Response

(The authors gave the same response as above.)

Round 2
Reviewer 1 Report
The authors have adequately addressed all my comments. I recommend acceptance for publication.